# Interaction, Insensitivity and Thermal Conductivity of CL-20/TNT-Based Polymer-Bonded Explosives through Molecular Dynamics Simulation

**DOI:** 10.3390/ijms241512067

**Published:** 2023-07-27

**Authors:** Shenshen Li, Qiaoli Li, Jijun Xiao

**Affiliations:** Molecules and Materials Computation Institute, School of Chemistry and Chemical Engineering, Nanjing University of Science and Technology, Nanjing 210094, China; lishenshen91@163.com (S.L.); liqiaoli720@163.com (Q.L.)

**Keywords:** molecular dynamics simulation, PBXs, interaction, thermal conductivity, insensitivity

## Abstract

Binders mixed with explosives to form polymer-bonded explosives (PBXs) can reduce the sensitivity of the base explosive by improving interfacial interactions. The interface formed between the binder and matrix explosive also affects the thermal conductivity. Low thermal conductivity may result in localized heat concentration inside the PBXs, causing the detonation of the explosive. To investigate the binder–explosive interfacial interactions and thermal conductivity, PBXs with polyurethane as the binder and 2,4,6,8,10,12-hexanitro-2,4,6,8,10,12-hexaazaisowurtzitane/2,4,6-trinitrotoluene (CL-20/TNT) co-crystal as the matrix explosive were investigated through molecular dynamics (MD) simulations and reverse non-equilibrium molecular dynamics (rNEMD) simulation. The analysis of the pair correlation function revealed that there are hydrogen bonding interactions between Estane5703 and CL-20/TNT. The length of the trigger bonds was adopted as a theoretical criterion of sensitivity, and the effect of polymer binders on the sensibility of PBXs was correlated by analyzing the interfacial trigger bonds and internal trigger bonds of PBXs for the first time. The results indicated that the decrease in sensitivity of CL-20/TNT mainly comes from the CL-20/TNT contact with Estane5703. Therefore, the sensitivity of CL-20/TNT-based PBXs can be further reduced by increasing the contact area between CL-20/TNT and Estane5703. The thermal conductivity of PBXs composed of Estane5703 and CL-20/TNT (0 0 1), (0 1 0) and (1 0 0) crystal planes, respectively, were calculated through rNEMD simulations, and the results showed that only the addition of Estane5703 to the (1 0 0) crystal plane can improve the thermal conductivity of PBX100.

## 1. Introduction

In the field of energetic materials, unifying the contradictory properties of high energy and low sensitivity has always been a research hotspot [1]. However, most current single-compound explosives cannot meet the requirements of high energy and low sensitivity at the same time, which seriously restricts their development and application. For example, 2,4,6,8,10,12-hexanitro-2,4,6,8,10,12-hexaazaisowurtzitane (CL-20) is the most famous high-energy-density compound currently in practical use, featuring high density, high detonation velocity and favorable oxygen balance [2,3]. However, CL-20 cannot meet the safety requirements of modern warfare and new weapons sufficiently because of the relatively high sensitivity [4,5]. With the development of co-crystal explosives in recent years, high energy and low sensitivity have been largely unified in them [6]. A co-crystal is a multi-component molecular crystal with a fixed stoichiometric ratio formed by two or more neutral molecules under the action of intermolecular non-covalent bonds (hydrogen bonds, van der Waals, π-π conjugation, etc.). By forming a co-crystal, the oxygen balance of some explosives can be effectively improved, the detonation speed and pressure are increased and the safety is also guaranteed [7]. Among them, the more typical CL-20/TNT co-crystal explosive not only has the properties of the high energy density of CL-20, but also has the insensitivity characteristics of trinitrotoluene (TNT) [8,9,10,11].

The trigger bond refers to the chemical bond that is most likely to be broken in the molecular structure of the energetic material [12]. For energetic compounds, there is a criterion that can theoretically judge the relative sensitivity [13]. It is generally believed that the trigger bond length is related to the sensitivity of the energetic material. According to the principle of smallest bond order (PBSO), based on quantum chemical calculation, for a series of energetic compounds, a smaller bond order of trigger bonds in molecular terms means the compound is more sensitive [14,15]. This principle has been used extensively in the prediction of impact sensitivity for various types of energetic compounds. Usually, chemical bond strength can be characterized by bond order and bond length in the molecule. But, for a certain bond type, classical MD simulation can only give a particular bond order, which is the same, and cannot provide electronic structure. However, MD simulation can provide statistical distribution of bond length. Thus, it is suitable to evaluate sensitivity based on the molecular structure parameter bond length, obtained through MD simulation.

In practical applications, explosives are used mixed with polymers. The combination of explosives with polymeric binders to form polymer-bonded explosives (PBXs) is an important advancement in high explosives science, offering improved safety and reliability, while maintaining performance [16,17]. Based on these advantages, PBXs are widely applied in many defense and economic scopes. Theoretical calculation studies on the sensitivity reduction effect of polymer binders in PBXs mostly focus on the interaction with explosive crystals and their mechanical properties [18,19,20]. However, researchers have found that the addition of some binders such as polycarbonate, polysulfone, polyester, etc. can increase the sensitivity of explosives [21]. This may be because the interaction force between these binders and explosives is small, the contact is not close enough or the addition of binders leads to a decrease in the thermal conductivity of PBXs compared to elemental explosives. In addition, the formation of the interface will affect the thermal conductivity of the PBXs [22,23,24]. The decrease in thermal conductivity will cause the local temperature of PBXs to rise, forming a hot spot, which can cause an explosion if the energy of such a hot spot is too high [25]. Therefore, molecular dynamics simulation to study the interface between the explosive crystal and the binder in PBXs and the thermal conductivity has theoretical significance to fill the gap.

In this paper, referring to polyurethane (Estane5703) as a high polymer binder in the formulation of PBXs [26], three PBX models were constructed by merging Estane5703 with CL-20/TNT crystal planes (0 0 1), (0 1 0) and (1 0 0), respectively. Using a molecular dynamics (MD) simulation method, the interaction between the CL-20/TNT crystal planes and the polymer binder was investigated through the calculation of the binding energy between different crystal planes and the polymer binder. Then, the pair correlation function was used to further analyze the interaction and the stacking structure of the interface. Through the calculation and comparison of the length of the trigger bonds at the interface and inside the CL-20/TNT co-crystal, it was found that adding a polymer binder could affect the length of the trigger bonds of the CL-20/TNT co-crystal at the interface. Using a reverse non-equilibrium molecular dynamics (rNEMD) simulation method [27], the thermal conductivity of all PBXs and their corresponding CL-20/TNT co-crystal models were calculated and compared. The mechanism of the effect of polymer binder on the sensitivity of PBXs was revealed from the perspective of thermal conductivity.

## 2. Results and Discussion

### 2.1. Glass Transition Temperature and Force Field Validation

When using molecular dynamics to simulate polymers, the glass transition temperature is usually used to verify the applicability of the equilibrium configuration and force field [28]. The glass transition is an important mechanical state transition phenomenon of polymer materials, and the corresponding glass transition temperature *T*_g_ is one of the important characteristic temperatures of polymer materials. The glass transition refers to the transition of a polymer between a glassy state and a highly elastic state. As the chain segments move when the glass transition occurs, the degree of change in the volume of the polymer will change. At the same time, due to the movement of the chain segments, the van der Waals force between the chain segments should change considerably around the glass transition temperature. The Estane5703 equilibrium structure obtained through MD simulation was subjected to 200 K to 500 K heating NPT simulation under atmospheric pressure conditions, and the simulation time was 5 ns. Thus, the curve of van der Waals energy with increasing temperature can be obtained, as shown in Figure 1a. In addition, referring to the most commonly used methods in molecular dynamics, the computation of the specific volume–temperature properties of Estane5703 is shown in Figure 1b. The Estane5703 cells were cooled from 450 K to 50 k in 50 k increments, and run equilibration for 5 ns at each temperature. All processes for volume equilibration were performed under NPT conditions.

The glass transition temperatures obtained from the van der Waals–temperature curve and volume–temperature curve were 255.7 K and 257.3 K, respectively, while its experimental value was 250 K [29,30,31,32], indicating that the selected force field is applicable. The effectiveness described by van der Waals energy had a direct impact on the investigation of the interaction between the binder and co-crystal. In addition, the thermal conductivity of Estane5703 calculated under the COMPASS force field was 0.196 W·K^−1^·m^−1^, which was close to the experimental value of 0.21 W·K^−1^·m^−1^. The accuracy of the thermal conductivity of the binder was also directly related to the effectiveness of the thermal conductivity evaluation of PBXs. The molecular dynamics simulation results of the density, thermal conductivity and glass transition temperature of Estane5703 were highly consistent with the experimental values, as shown in Table 1, demonstrating the applicability of the simulation method and force field to Estane5703.

### 2.2. Interface Interaction

The binding energy was defined as the negative value of the intermolecular interaction energy (*E*_inter_). The interaction energy between the explosive and the polymer binder was calculated by subtracting the individual component energy in the system from the total energy of the whole system. Thus, the binding energy can be expressed as follows [2]:*E*_bind_ = −*E*_inter_ = −(*E*_total_ − *E*_CL-20/TNT_ − *E*_Estane5703_)(1)
where *E*_bind_ and *E*_inter_ represent the binding energy and interaction energy, respectively; *E*_total_ represents the total energy of PBXs; *E*_CL-20/TNT_ represents the energy of CL-20/TNT matrix explosives and *E*_Estane5703_ represents the energy of Estane5703. For PBXs, the binding energy indicated the strength of the interaction between the polymer binder and the base explosive. In this paper, binding energy was used to reflect the thermal stability of energetic systems. Table 2 shows the binding energy and the energy of the corresponding components between the crystal faces of three different co-crystal explosives and Estane5703. From Table 2, it can be seen that PBX100 has the largest binding energy value and PBX001 has the smallest binding energy value. Due to differences in interfacial area, *E*_bind_′ was used to represent the binding energy per unit interfacial area, where *E*_bind_′ is the *E*_bind_ of the unit interfacial area. By comparing *E*_bind_′, it was discovered that the thermal stability follows the order PBX100 > PBX010 > PBX001. The above showed that the interfacial interactions between Estane5703 and the CL-20/TNT crystal planes differed significantly, and the largest value of *E*_bind_ and *E*_bind_′ indicated that the interactions between Estane5703 and the (1 0 0) crystal plane were the strongest.

### 2.3. Pair Correlation Function

The pair correlation function (PCF) describes the probability number density *g*(r) with the distance of the reference particle, representing the possibility of finding a certain particle at the distance *r* from the reference particle. Therefore, by analyzing the *g*(r) of different atom pairs at the Estane5703 and CL-20/TNT interface, an insight into the material structure could be gained through the revealed local spatial ordering. Because the interaction between particles decreased sharply when the distance between particles was longer, only the atoms closer to the interface between Estane5703 and CL-20/TNT needed to be considered. The pair correlation function of the interfacial structure atom pair of PBX001, PBX010 and PBX100 are shown in Figure 2. For convenience, the hydrogen atom, oxygen atom and nitrogen atom in CL-20/TNT are denoted as H_1_, O_1_ and N_1_, respectively. The hydrogen atom and oxygen atom in Estane5703 are denoted as H_2_ and O_2_, respectively.

Non-bonding interactions between atoms include hydrogen bonds, van der Waals (vdW) and electrostatic interactions. In general, the distance range for hydrogen bonding interactions was 2.0–3.1 Å and that of the vdW and electrostatic interactions was usually 3.1–5.0 Å. When the distance between two atoms was farther than 5.0 Å, the interaction was quite weak. It can be seen from Figure 2 that in the hydrogen bond range, the PCF curves of O_1_–H_2_ and H_1_–O_2_ all displayed comparatively high peaks, indicating that the hydrogen bonds exist in O_1_–H_2_ pairs and H_1_–O_2_ pairs for PBX001, PBX010 and PBX100. Figure 3 displays a visual representation of the hydrogen bonding interactions of O_1_–H_2_ in PBX100 as an example. Although there were hydrogen bond interactions in the interface structure composed of the three crystal planes of Estane5703 and CL-20/TNT co-crystal, the peaks in the hydrogen bond range varied in intensity. For PBX001, in the hydrogen bond range, it could be found that the *g*(r) value of O_1_–H_2_ was significantly less than that of H_1_–O_2_. Similarly, the *g*(r) value of O_1_-H_2_ was larger than that of O_1_–H_2_ in PBX010, but the *g*(r) values of O_1_–H_2_ and H_1_–O_2_ were almost the same in PBX100. For N_1_−H_2_ in Figure 2, the curve had no peak for hydrogen bond interactions, implying that only vdW and electrostatic interactions existed between N_1_−H_2_ pairs.

For details, the integral areas of the PCF curves were calculated. The difference in the interaction between the three atoms pairs can be clearly compared by integrating the PCF curve, as shown in Table 3. The value of the integrated area of the correlation function can reflect the magnitude of the interaction strength [35]. It can be concluded from Table 3 that the strength of the hydrogen bonds in the three interface structures is less than that of the van der Waals interaction, which indicates that there are hydrogen bond interactions in the interface structure but the van der Waals interaction is greater than the hydrogen bond. From Table 3, it can also be seen that the integral areas for the hydrogen bonds range in PBX001 were the smallest. The integration area of O_1_–H_2_ was the largest in PBX010 and the integration area of H_1_–O_2_ was the largest in PBX100. For distance in the hydrogen bond range, PBX010 had the smallest integral area, which also coincided with the smallest value of its binding energy.

### 2.4. N-NO_2_ Trigger Bond Length

Compared with C-NO_2_, N-NO_2_ is a more easily broken trigger bond. Additionally, it is well known that CL-20 is more sensitive than TNT, and the CL-20 component in CL-20/TNT decomposes first in detonation. Therefore, the N-NO_2_ bond in CL-20 was chosen as the trigger bond in this work. The maximum trigger bond length (*L*_max_) and the average trigger bond length (*L*_ave_) were used to characterize the sensitivity of CL-20/TNT co-crystal-based PBXs. In addition, since the interaction of the polymer binder at the CL-20/TNT interface will cause the sensitivity of CL-20/TNT at the interface to be different from that of CL-20/TNT that is not in contact with the binder, the trigger bond of PBXs was divided into both interface and internal parts for analysis. Table 4 lists the maximum bond length *L*_max_ and the average bond length *L*_ave_ of N-NO_2_. 

It can been seen from Table 4 that the bond lengths of N-NO_2_ are all smaller than those of CL-20/TNT, for PBX001, PBX010 and PBX100, which indicates that the addition of the binder Estane5703 can make the sensitivity decrease. The *L*_ave_ at the interface of PBX001 and the *L*_max_ of PBX001 are essentially unchanged compared to CL-20/TNT, which shows that placing Estane5703 on the (0 0 1) crystal plane of CL-20/TNT has less effect on the sensitivity. After that, the *L*_max_ of PBX010 and PBX100 are smaller than the *L*_max_ of CL-20/TNT, and the *L*_ave_ of PBX010 and PBX100 are markedly smaller than the *L*_ave_ of CL-20/TNT, which indicates that the addition of the binder Estane5703 made the sensitivity decrease of PBX010 and PBX100 more obvious. In addition, it can be found that the *L*_max_ and *L*_ave_ at the interface are smaller than the internal *L*_max_ and *L*_ave_, which indicates that Estane5703 decreased the sensitivity of CL-20/TNT, mainly due to the decreased sensitivity of CL-20/TNT in contact with Estane5703. At the same time, this also shows that the sensitivity of CL-20/TNT can be reduced by increasing the contact area between the polymer binder Estane5703 and CL-20/TNT.

### 2.5. Thermal Conductivity

For energetic materials, if their thermal conductivity is low, the local temperature may rise quickly because of the localized heat concentration inside the PBXs, which may detonate the energetic materials. Figure 4 shows the thermal conductivity along the *c*-axis of PBXs with three different crystal planes and CL-20/TNT with three different crystal planes. Due to the different atomic arrangements in the three crystal planes (0 0 1), (0 1 0) and (1 0 0), the thermal conductivity of the CL-20/TNT co-crystal was also different. Among them, the thermal conductivity of the (0 0 1) crystal plane along the *c*-axis was the best. The thermal conductivity of PBX001 and PBX010 along the *c*-axis was lower than their corresponding CL-20/TNT co-crystals. The decreased thermal conductivity of PBX001 formed by adding the adhesive Estane5703 in (0 0 1) might have increased the sensitivity of CL-20/TNT, which is consistent with the conclusion from the PBX001 trigger bonds. The thermal conductivity of PBX010 formed by adding the adhesive Estane5703 in the (0 1 0) plane was almost unchanged compared with that of CL-20/TNT. The thermal conductivity of PBX100 was greater than CL-20/TNT, indicating that adding Estane5703 in the (1 0 0) crystal plane can improve the thermal conductivity of PBX100.

## 3. Modelling and Computational Methods

### 3.1. Amorphous Polymer Chain

The molecular structure of polyurethane (Estane5703) is shown in Figure 5a. This polymer is a copolymer composed of hard segments and soft segments. In this paper, the number of repeating units of hard segments was m = 1, the number of repeating units of soft segments was n = 4 and the end groups were saturated with H and CH_3_ to form the initial stage of Estane5703. The Monte Carlo method was used to generate twist angle in the chain to obtain the random conformation of the Estane5703 molecular chain. Then, six Estane5703 molecular chains were put into a periodic box with a low density, and a density-convergent Estane5703 amorphous model was obtained by using the high-low pressure dynamic simulation method [36]. This was the initial model of the polymer binder in this paper.

### 3.2. Models of CL-20/TNT and PBXs

Based on the data of CL-20/TNT co-crystal obtained using X-ray analysis [37], a (2 × 1 × 1) supercell model was built. Since the molecular arrangements of CL-20/TNT in the three crystal plane directions (0 0 1), (0 1 0) and (1 0 0) were quite different, these three crystal planes were selected to study the interfacial interaction and thermal conductivity of CL-20/TNT-based PBX. First, the CL-20/TNT supercell model was cut along the three crystal plane directions (0 0 1), (0 1 0) and (1 0 0), respectively. The *c* lattice heights of the obtained (0 0 1), (0 1 0) and (1 0 0) periodic boxes were 24.69 Å, 19.37 Å and 19.35 Å, respectively, and the surface areas, *a* × *b*, were 19.35 Å × 19.37 Å, 24.69 Å × 19.35 Å and 19.37 Å × 24.69 Å, respectively.

Taking the (0 0 1) crystal plane model as an example, the steps to establish a PBX model for the corresponding crystal plane were as follows. An Estane5703 molecular chain was chosen from the initial model of the polymer binder and put in a period box with the same size as the (0 0 1) crystal plane model. Then, keeping the surface (*a* × *b*) unchanged, the height of the *c*-axis was adjusted gradually until the theoretical density of Estane5703 was reached; each adjustment needed the molecular dynamics simulation running to equilibrium state. After that, the (0 0 1) crystal plane model was merged at the two ends of the Estane5703 period box perpendicular to the *c*-axis, respectively. Therefore, the PBX001 structure was obtained, containing 1994 atoms, as shown in Figure 6. The (0 1 0) and (1 0 0) crystal plane structures obtained through the same method are denoted as PBX010 and PBX100, respectively. In this way, the weight percentages of the binders in the PBXs are all about 5.2%, within the general control range of polymer binder content in explosives [21].

The above models for CL-20/TNT co-crystal and CL-20/TNT-based PBXs were allowed to evolve dynamically in isothermal–isobaric (NPT) ensemble with Andersen temperature control using the stochastic collision method and Parrinello–Rahman [38] pressure control, fully relaxing all cell parameters at atmospheric pressure. The van der Waals interactions were truncated at 12.5 Å with long-range tail correction, and the electrostatic interactions were calculated via the standard Ewald summation [39]. The equations of motion were integrated with a step of 1 fs. An equilibration run was performed for 5 ns. After the equilibration run, production runs of 1 ns were performed, during which data were collected with 10 fs sampling intervals for analysis. In this paper, all simulations were conducted utilizing a COMPASS (condensed-phase optimized molecular potentials for atomistic simulation studies) force field [40], which was suitable for MD simulation of the condensed phase, especially for nitramine explosives [16,41,42,43,44].

### 3.3. Models and Methods for Thermal Conductivity

The thermal conductivity of PBXs was calculated through a reverse non-equilibrium MD (rNEMD) [45,46], whose advantage was fast convergence of the temperature gradient in a non-equilibrium stable system. For the rNEMD, we used the Muller-Plathe algorithm to exchange kinetic energy between atoms in the hot region and cold region.

The first step was to establish an appropriate thermal conductivity model, and then apply heat flux to the model in one direction to generate a temperature gradient in the same direction. The thermal conductivity can be further obtained by calculating the heat flux along the temperature gradient. Regarding the thermal conductivity model, for the PBX model, in order to ensure that the heat flux completely passed through the interface between CL-20/TNT and Estane5703, two identical PBX models were merged into their thermal conductivity model along the pseudo-flow direction of heat, that is, the *c*-axis direction. Figure 7 displays the thermal conductivity model of PBX001 as an example where the heat flux is along the *c*-axis from the hot region to the cold region.

The simulated CL-20/TNT or PBX system was divided into 50 equal slices along the direction of heat flux (*c*-axis in this paper), where the middle slice is the heat source layer (hot region) and the first slices of the system are the heat sink layer (cold region). The coldest atom in the hot region and the hottest atom in the cold region were paired up and their velocities were exchanged [47]. The exchange could effectively swap kinetic energies of atoms, assuming their masses were the same. If the masses were different, a velocity swap was performed relative to the motion of the center of mass of the atoms to conserve kinetic energy. Thus, the energy could be transferred from the hot region to the cold region, and a temperature gradient could be induced in the simulated box. The heat flux can be obtained by calculating the exchanged amount of kinetic energy according to the following formula:(2)$$J=\frac{\sum_{Nswap}\frac{1}{2}\left(mv_{h}^{2}-mv_{c}^{2}\right)}{t_{swap}}$$
where *N_swap_* is the number of swaps to perform for energy conversion; *t*_swap_ is the intervals to perform kinetic energy exchange; *m* is the mass of the exchanged atoms and *v*_h_ and *v*_c_ are the speeds of the hottest and coldest atoms, respectively. Then, the thermal conductivity can be calculated using the Fourier formula:(3)$$\kappa =\frac{J}{2A\partial T/\partial z}$$
in which *A* is the cross-sectional area in the direction of heat flux and *∂T*⁄*∂z* is the derivative of the system temperature with respect to the direction of heat flux. Due to the exchange of the velocity of the related atoms in the heat source layer and the heat sink layer, the temperature gradient in the system could be generated. The temperature gradient of the system was obtained by calculating the temperature of 50 equal parts of the uniform lamella, as shown in Figure 8. The temperature of each layer can be calculated using the following formula:(4)$$T_{i}\left(slab\right)=\frac{2}{3Nk_{B}}\sum_{j}\frac{p_{j}^{2}}{2m}$$
where *N* is the number of atoms in the slice (on average, each slice contains 80 atoms); *k*_B_ is Boltzmann’s constant; *p*_j_ is the momentum of the atom and *m* is the mass of the atom.

In this thermal conductivity simulation, the equations of motion were integrated with a step of 1 fs, and the temperature and pressure were controlled via the Nose–Hoover thermostat and barostat [48,49]. Van der Waals interactions were truncated atom pairs with interatomic distances greater than 12.5 Å, coupled with long-range tail correction. And electrostatic interactions were processed using standard Ewald summation. The simulation systems were first equilibrated under the NPT ensemble at 300 K and atmospheric pressure run for 2 ns. Then, the exchange of kinetic energies was performed for 2 ns. The temperature profile was computed by averaging events in a time interval of 20 ps. Five independent equilibration and NEMD simulations were performed for each crystal orientation. These computations were all carried out using the software LAMMPS (5 June 2019) (Large-scale Atomic/Molecular Massively Parallel Simulator) [50].

## 4. Conclusions

In this paper, the interface interaction and sensitivity of PBXs formed by adding the polymer binder Estane5703 in the (0 0 1), (0 1 0) and (1 0 0) crystal planes of CL-20/TNT were studied through molecular dynamics simulations. The simulations involved binding energy calculation, PCF analysis, bond length of the N–NO_2_ trigger bond and thermal conductivity.

From the calculated binding energies, the values of *E*_bind_ and *E*_bind’_ between Estane5703 and CL-20/TNT for PBX100 were the largest. It was found that PBX100 has the best thermal stability, which means that Estane5703 combined has the best thermal stability in the (1 0 0) crystal plane. Additionally, the PCF analysis of atom pairs in the interfacial structure indicated that there exists hydrogen bond between CL-20/TNT and Estane5703 in all three crystal planes. The hydrogen bonds mainly came from H_1_−O_2_ and O_1_−H_2_, but there were far fewer hydrogen bonds present in the N_1_−H_2_ atomic pairs. The PCF also indicated that the number of strong vdW and electrostatic interacting atomic pairs is much greater than the number of hydrogen bond interacting atomic pairs. By analyzing the length of the N-NO_2_ trigger bond, it was found that the *L*_max_ and *L*_ave_ at the interface are both smaller than the internal *L*_max_ and *L*_ave_, indicating that the main reason for the Estane5703 reducing the sensitivity of CL-20/TNT is that it reduces the contact with Estane5703. Therefore, the sensitivity can be reduced by increasing the contact area of the binder Estane5703 with the CL-20/TNT co-crystal. The thermal conductivity of PBXs showed that merging Estane5703 in the (0 0 1) and (0 1 0) crystal plane can lead to a decrease in the thermal conductivity of CL-20/TNT in the corresponding crystal planes, and only merging Estane5703 in the (1 0 0) crystal plane can improve the thermal conductivity of CL-20/TNT in the corresponding crystal planes. This research work is helpful to deeply understand the interaction between the binder and the explosive in polymer-bonded explosives, and to further reveal the mechanism of the reduction in the sensitivity of the explosive in polymer-bonded explosives.

## Figures and Tables

**Figure 1 ijms-24-12067-f001:**
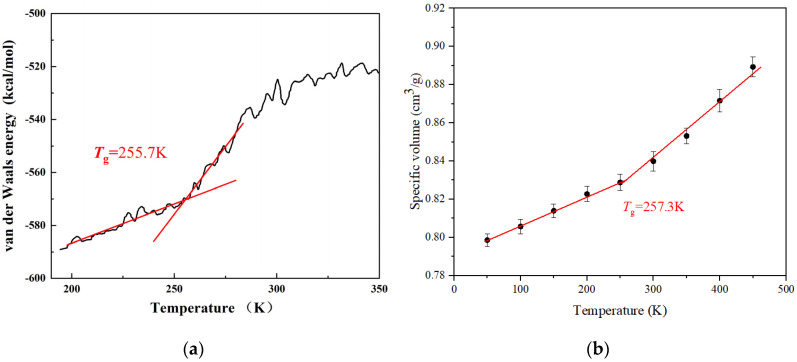
(**a**) Van der Waals energy vs. temperature for Estane5703 and (**b**) specific volume vs. temperature for Estane5703. The glass transition temperatures were identified as the intersection of the red lines (fitting line).

**Figure 2 ijms-24-12067-f002:**
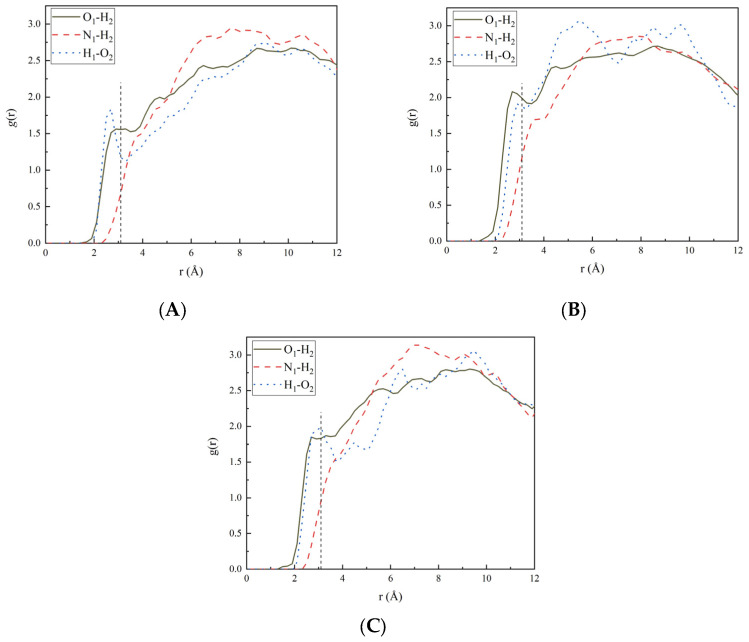
PCF for atom pairs in CL−20/TNT-based PBXs. (**A**) PBX001, (**B**) PBX010, (**C**) PBX100.

**Figure 3 ijms-24-12067-f003:**
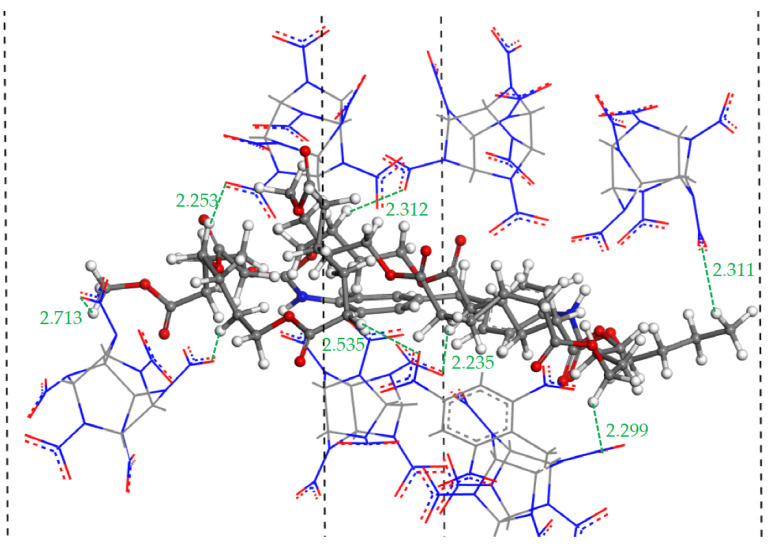
Illustration of hydrogen bonds of O_1_−H_2_ in PBX100: O_1_ atoms in the co-crystal (red line model) and H_2_ atoms in Estane5703 (white ball).

**Figure 4 ijms-24-12067-f004:**
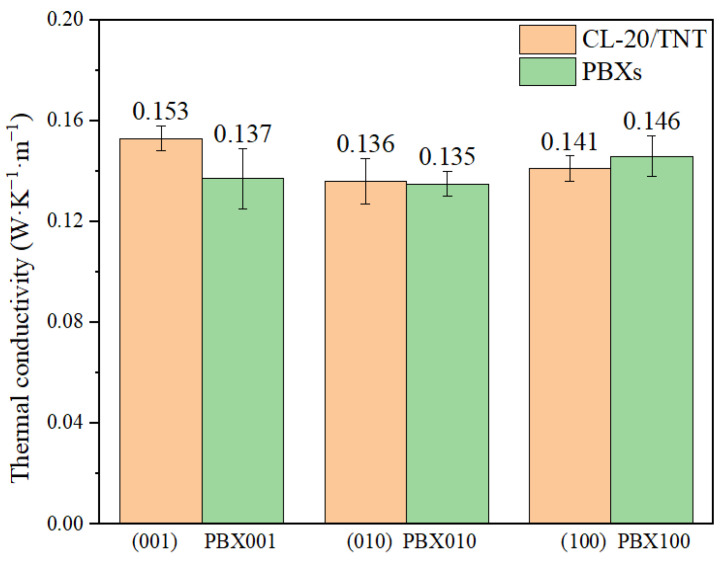
The thermal conductivity of CL-20/TNT co-crystal and PBXs with three different crystal planes.

**Figure 5 ijms-24-12067-f005:**
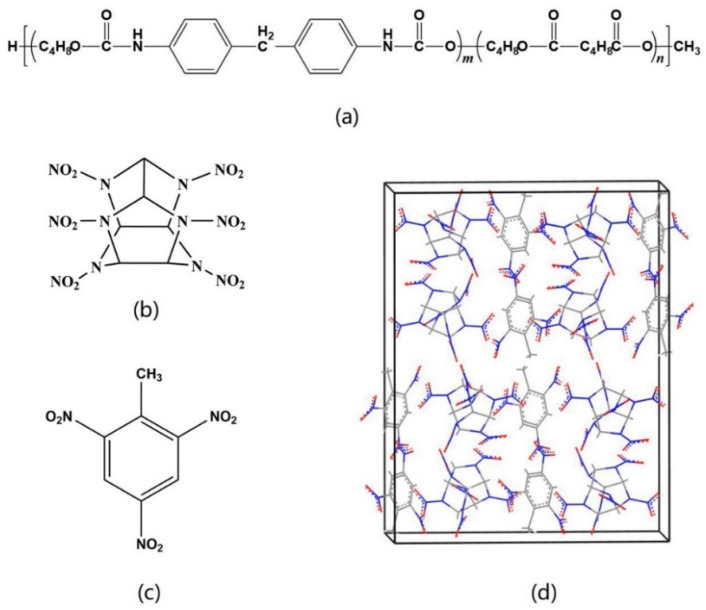
Molecular structures of (**a**) Estane5703, (**b**) CL-20 and (**c**) TNT; (**d**) co-crystal structure of CL-20/TNT.

**Figure 6 ijms-24-12067-f006:**
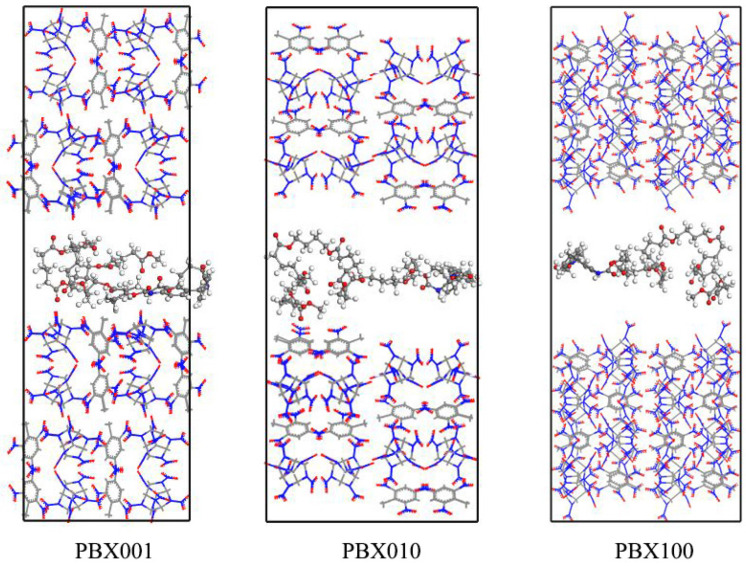
PBXs structures of different crystal planes. The CL-20/TNT co-crystal in stick model and Estane5703 in ball–stick.

**Figure 7 ijms-24-12067-f007:**
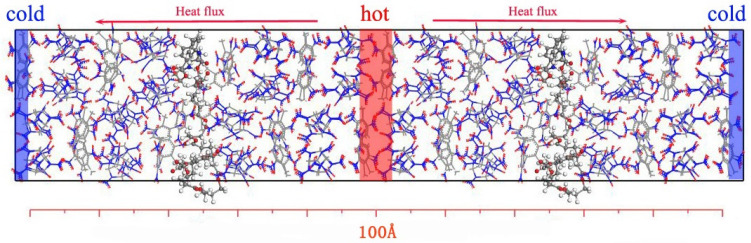
Schematic picture for heat flux setting in PBX001.

**Figure 8 ijms-24-12067-f008:**
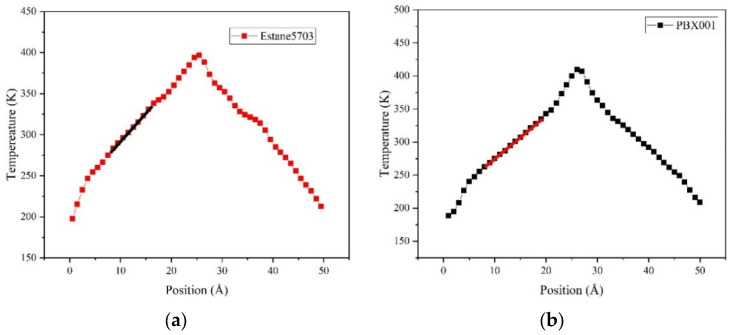
The temperature profile of (**a**) Estane5703 and (**b**) PBX001 (the linear fitting region is labelled with a line segment).

**Table 1 ijms-24-12067-t001:** Density, thermal conductivity and glass transition temperature of Estane5703.

Polymer	Density (g·cm^−3^)		Thermal Conductivity (W·K^−1^·m^−1^)		*T*_g_ (K)		
	MD	Experiment	MD	Experiment	MD ^1^	MD ^2^	Experiment
Estane5703	1.19	1.21 [33]	0.20	0.21 [34]	255.7	257.3	~250 [29,32]

^1^ calculated from Figure 1a; ^2^ calculated from Figure 1b.

**Table 2 ijms-24-12067-t002:** Binding energy, total energy and energy of different PBXs.

Model	*E*_bind_ (kcal·mol^−1^)	*E*_total_ (kcal·mol^−1^)	*E*_Estane5703_ (kcal·mol^−1^)	*E*_CL-20TNT_ (kcal·mol^−1^)	*E*_bind_’ (kcal·mol^−1^·nm^−2^)
PBX001	159.22	−8558.08	−72.81	−8326.05	43.09
	(7.13)	(30.64)	(9.07)	(31.27)	(1.92)
PBX010	178.16	−8496.69	−68.04	−8250.49	44.69
	(4.47)	(30.82)	(9.16)	(28.65)	(1.10)
PBX100	193.97	−8505.28	−61.96	−8249.35	50.07
	(6.75)	(29.99)	(7.82)	(31.93)	(1.73)

The corresponding deviations are listed in brackets.

**Table 3 ijms-24-12067-t003:** Integral areas of PCF curves for PBXs.

Model	Distance (Å)	O_1_–H_2_	H_1_–O_2_	N_1_–H_2_
PBX001	2.0–3.1	1.25	1.34	0.20
	3.1–5.0	3.12	2.44	2.70
PBX010	2.0–3.1	1.77	1.20	0.41
	3.1–5.0	3.92	4.20	3.18
PBX100	2.0–3.1	1.54	1.35	0.31
	3.1–5.0	3.66	3.04	3.01

**Table 4 ijms-24-12067-t004:** The trigger bond (N-NO_2_) lengths (Å) of CL-20 for the co-crystal and PBXs.

Model		*L* _ave_	*L* _max_
PBX001	interfacial	1.4115 (0.0325) −0.56	1.5423 (0.0012) −0.59
	internal	1.4096 (0.0320) −1.91	1.5475 (0.0009) −0.26
PBX010	interfacial	1.4089 (0.0324) −2.41	1.5363 (0.0016) −0.98
	internal	1.4102 (0.0319) −1.48	1.5433 (0.0017) −0.53
PBX100	interfacial	1.4095 (0.0316) −1.98	1.5283 (0.0011) −1.49
	internal	1.4107 (0.0320) −1.13	1.5473 (0.0012) −0.27
CL-20/TNT		1.4123 (0.0319)	1.5515 (0.0012)

The corresponding deviations are listed in parentheses. The bond elongations are given in permillage (‰), where the bond length in CL-20/TNT is used as a reference.

## Data Availability

All data are contained within the article.

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
