# Peer review of "Interaction, Insensitivity and Thermal Conductivity of CL-20/TNT-Based Polymer-Bonded Explosives through Molecular Dynamics Simulation"

_ijms, 2023, doi:10.3390/ijms241512067_

Round 1

Reviewer 1 Report

(1) The authors should consider the choice of force field. The  COMPASS  was used in this study, using other force fields will lead to the consistent conclusion or not?

(2) The size of 100 A in the direction of heat flux was used, the size effect should be verified.

(3)"The thermal conductivity of PBXs showed that adding Estane5703 on the (1 0 0) crystal plane can improve the thermal conductivity of PBX100. "  Is this statement reasonable? The author should  talk more about the interface heat transfer.  

The English is well

Reviewer 2 Report

The paper reports an original study of the interface between the high energy material (CL-20/TNT mixture) and polymeric binder (Elastane 5703) as the components of a polymer-bounded explosive.

The authors employ a combination of molecular dynamics simulations two evaluate the two key parameters for this application – thermal conductivity (directly - through NEMD simulation) and sensitivity (indirectly through analysis of trigger bonds).

The contribution falls into the scope of the journal and would definitely be of interest both for chemists involved in high energy materials research as well as for molecular simulation experts.

The paper is in general well written, the employed methods are adequate for the task and with a few exceptions well described.

A few moments require correction before the contribution can be accepted for publication:

A. Experimental setup and results:

            1. My main concern is that the simulated system appears to be rather small – just 2x1x1 supercell for structure equilibration and, apparently, twice that size for NEMD simulation. How many molecules and atoms did simulated supercells contain? How many independent equilibration and NEMD simulations were performed for each crystal orientation? How many atoms on average contained slices for temperature averaging in NEMD simulations? Also, was the temperature values for each slice averaged across any number of time steps? This information should be provided to evaluate the validity of the reported results. Furthermore, the error bars in the figure 4 indicating variation of local temperature across the sampling time interval (and independent simulations, if there were any) should be drawn to evaluate the quality of the calculated data.

2. The authors derive the conclusion regarding the effect of binder on sensitivity of compound explosive by analyzing the length of trigger bonds on the interface and inside the blocks of high energy component. However, the difference in the bond lengths reported in the Table 4 are quite small relative to the estimated deviations (of the order ~0.01). For example, on the lines 317-318 the authors state “the Lmax and Lave of PBX010 and PBX100 are all smaller than CL-20/TNT, which indicates that the addition of binder Estane5703 makes the sensitivity of PBX010 and PBX100 decrease”. However, from the Table 4 we see (if rounded to the second digit) that for PBX010 Lmax(interface) = 1.49+/-0.02 Lmax(internal) = 1.50+/-0.02.
Given the previous concerns that the simulation system is rather small, either more data is needed to reduce deviation below ~0.001 (ideally – simulation of larger supercell, but several independent simulations or sampling during longer time could suffice) or a very good explanation from the authors, why they believe their conclusions can be derived from such data.

            3. The authors use the timestep of 1 fs. For all-atom simulations including explicit hydrogen atoms it is customary to use 0.5 fs. A comment is needed in the methods section on why authors believe such time step is enough to satisfactory describe hydrogen dynamics. This is especially relevant since hydrogen bonding plays an important role in interface bonding.

            4. Lines 123-127 the authors state that the equilibration run was 5 ns “and should be extended” – so after all, how long in simulation time was needed to achieve equilibrium?

            5. The authors use Andersen thermostat and Parrinello-Rahman barostat for structure equilibration and Nose-Hoover thermo- and barostats for NEMD simulations. A comment from the authors explaining such particular choice of methods would be helpful.

B. Organization of the paper:

            1. From the very beginning the paper uses terms “sensitivity” and “trigger bonds”, yet their explanation first appears on page 10. It is desirable to move the first paragraph of the section 3.4 into the introduction section to give a general reader some clarity about these terms and their importance for the subject. Also for a general reader somewhat more explicit definition of what is sensitivity of an explosive would be helpful.

            2. The paper features a separate “Modelling and Computational Methods” section, yet the “Results” section begins with describing technical details of evaluation of the glass transition temperature. It is advisable to include them into the previous section and focus on the discussion of the results.

While the paper is in general well and clearly written a few moments require correction or change:

Most importantly: the argument on the lines 299-303 on the use of bond lengths as criteria of sensitivity must be revised. Namely, on the line 299 the authors state: “Classical MD simulation… cannot give the bond order data”. This statement is false. In classical MD the strength and equilibrium length parameters of each covalent bond are assigned according to known bond types and this presumes a particular bond order. In LAMMPS it is even possible to extract the elastic energy of each bond and, provided the energy required for bond breaking, to identify which particular bond will break first.

I understand the idea of the authors, that watching the lengths of the critical bonds is easier and serves as a good indicator of which bonds are closer to breaking point. But it would be better just to say so in the text. Also, from this point of view, the discussion would be clearer if instead of absolute lengths of the bonds the authors reported bond elongations (dl=l(t)-l0, l0 being the equilibrium bond length) or even relative value (dl/l0) in %. The absolute values of bond length do not tell much to a general reader, while larger elongation clearly indicates large strain.

Other desired improvements:

line 46: “explosives appear in a state” better “explosives are used as…”

line 60: “molecular dynamics simulation calculation” – “calculation” is redundant.

line 61: “the interface interaction of the interface” – one interface is enough.

line 63: “referring to the general content of polyurethane (Estane5703) as a high polymer binder” – “the general content of” is redundant.

lines 81-82: “In this paper, the number of segments in the hard segment was m = 1, the number of segments in the soft segment was n = 4” – here the word “segments” are obviously used in two different meanings without explanation. Revise the sentence.

line 86: “period box” – “periodic box”

line 91: “Molecule” – better “Molecular structures”

line 112: “within the generally controlled range.” – it is unclear what the authors mean by “generally controlled range”.

line 141 “fluxing” – just “flux”

line 154 “speed swap” – should be “velocity swap”

lines 320-321: “which indicates that the main reason for the Estane5703 to reduce the sensitivity of CL-20/TNT is reduced the sensitivity of CL-20/TNT contacted with Estane5703.” Consider revising. The idea can be understood after a long and careful reading for a few times, but on the first glance appears like “the main reason for effect A is the effect A”.

line 329: “Thermal conductivity κ was an important index to characterize the thermal conductivity of materials.” – a completely redundant sentence.
